# Photocatalysis in Water-Soluble Supramolecular Metal Organic Complex

**DOI:** 10.3390/molecules28104068

**Published:** 2023-05-12

**Authors:** Dongfeng Hong, Linlin Shi, Xianghui Liu, Huiyuan Ya, Xin Han

**Affiliations:** 1College of Food and Drug, Henan Functional Cosmetics Engineering & Technology Research Center, Luoyang Normal University, Luoyang 471934, China; 2College of Chemistry, Zhengzhou University, Zhengzhou 450001, China

**Keywords:** metal-organic cages, design and synthesis, catalysis, metal complexes, photosynthesis

## Abstract

As an emerging subset of organic complexes, metal complexes have garnered considerable attention owing to their outstanding structures, properties, and applications. In this content, metal-organic cages (MOCs) with defined shapes and sizes provide internal spaces to isolate water for guest molecules, which can be selectively captured, isolated, and released to achieve control over chemical reactions. Complex supramolecules are constructed by simulating the self-assembly behavior of the molecules or structures in nature. For this purpose, massive amounts of cavity-containing supramolecules, such as metal-organic cages (MOCs), have been extensively explored for a large variety of reactions with a high degree of reactivity and selectivity. Because sunlight and water are necessary for the process of photosynthesis, water-soluble metal-organic cages (WSMOCs) are ideal platforms for photo-responsive stimulation and photo-mediated transformation by simulating photosynthesis due to their defined sizes, shapes, and high modularization of metal centers and ligands. Therefore, the design and synthesis of WSMOCs with uncommon geometries embedded with functional building units is of immense importance for artificial photo-responsive stimulation and photo-mediated transformation. In this review, we introduce the general synthetic strategies of WSMOCs and their applications in this sparking field.

## 1. Introduction

Photosynthesis is a process of harvesting solar light and converting it into chemical energy. It can turn inorganic matter into organic matter and is the most important chemical reaction on earth [1,2,3]. In this process, the excitation energy migrates from sunlight-absorbing chlorophyll, which is embedded in light-harvesting complexes known as antenna proteins, to the reaction center and then converts into chemical energy [4,5]. In the chemistry world, photoredox catalysis is widely used in the field of organic synthesis because of its ability to construct new species by mimicking photosynthesis. In particular, as an ecological and environmental protection technology [6,7], photocatalysis has been widely used in water oxidation [8,9,10,11], CO_2_ reduction [12,13,14], H_2_S oxidation [15,16], etc. It is worth noting that initiating photocatalysis generally requires organic dyes as photocatalysts to convert visible light into chemical energy under mild conditions. The organic dye molecules irradiated with visible light can absorb light and transfer energy to substrates by various mechanisms, leading to activation of the reactions [17,18,19,20].

Similar to the nano-environments in natural systems, the hydrophobic cavities of water-soluble metal-organic cages (WSMOCs) with defined shapes and sizes provide internal spaces to isolate water for guest molecules, which can be selectively captured, isolated, and released to achieve control over chemical reactions [21,22]. WSMOCs are known to provide various bond geometries, orientations, and strengths and show excellent catalytic performance due to the diversity of ligands, mental centers, coordination sites, and configurations [23,24,25]. To date, the reported mediated reactions include the Diels-Alder reaction [26,27,28], Kemp elimination [29,30], Knoevenagel condensation [31,32,33], pericyclic reactions (e.g., [2 + 2]/[4 + 2] cycloaddition reactions) [34,35,36], etc., wherein most of these reactions were mediated by the tail-anchored functional group and/or the confined space for enhancing the local concentration and pre-organizing the substrates [37,38,39]. As a result, the design and synthesis of WSMOC architectures have emerged as one of the most dynamic fields within supramolecular chemistry, not only because of their charming structures for mediating chemical reactions [40,41,42,43] but also as a result of their various applications in molecular recognition [44,45,46,47], biomimetics [48,49,50], guest sequestration [51,52], reactive species stabilization [53,54,55,56,57], drug delivery [58,59,60], membrane transportation [61,62,63], and so forth.

Recently, great progress has been achieved for photochemical reactions by using WSMOCs as photocatalysts, in which the photosensitizers are covalently incorporated into the cages as part of the backbones or as tail-anchored functional groups [64,65,66]. However, the reactions of photochemical molecules are difficult to control due to their essentially low reaction barriers and high activity intermediates upon excitation [67]. Additionally, implementing such a strategy would need to fulfill the following two criteria: (1) sufficiently large inner space to accommodate structurally complicated catalysts and substrates; (2) sufficiently strong driving force to ensure the encapsulation; (3) possession of light-absorbing functionalities; (4) consideration that the readily prepared organic dyes could be decorated to stable metallacage skeletons and their optical characteristics can be enhanced [68,69,70,71,72]. As a result, the design and construction of WSMOCs or WSMOC-based light-harvesting systems as nanoreactors to enable photocatalytic reactions in aqueous media have recently attracted increasing attention.

Although some successful approaches have been adopted for photocatalytic reactions in supramolecular architectures based on covalent organic frameworks (COFs) [73,74,75] and metal organic frameworks (MOFs) [76,77], such systems are easy to synthesize, but their poor solubility and stability in common solvents restrict their processability for further application. Supramolecular coordination complexes (SCCs) formed by ligand-driven self-assembly have the advantages of facile one-pot synthesis, good stability, and high solubility in common solvents, which provide an alternative hope in this regard. In similar fashion to the nanometer-scale compartments in living systems, WSCCs can isolate guest molecules and host chemical transformations. Therefore, these cages have a wide range of applications in biological, medical, environmental, and industrial fields, and this conforms to the development concept of green chemistry. In this review, we first briefly introduce synthetic methods for the construction of WSMOCs. Then, the WSMOCs with application in photocatalysis in recent years are also summarized. The unique microenvironments in the cavities of WSMOCs have been extensively used to encapsulate various guests by means of electrostatic, hydrophobic, and van der Waals interactions in the aqueous phase [78,79,80]. Therefore, we introduce the application of several different types of WSMOCs in photocatalytic reactions. It is worthy to note that the MOCs discussed in this review are only prepared and dissolved in mixed solvents containing at least 50% water. At the end, the development prospects of WSCCs in supramolecular photocatalysis are summarized and prospected.

## 2. Strategies to Prepare Water-Soluble Metal-Organic Cages

Despite key early work by the Fujita and Raymond groups on water-soluble cages, most cage work since has been in organic solutions. Some of the early coordination assemblies turned out to be soluble and stable in water due to the combination of charge and polar functional groups within their structures. In recent years, strategies for the synthesis of WSMOCs have been developed, which are depicted in Figure 1.

### 2.1. Metal Centers

A series of water-soluble metal-organic cages have been constructed in recent years since the first water-soluble quadrilateral metal ring was reported by the group of Fujita [81]. For example, in 1995, Fujita and co-workers successfully synthesized a water-soluble octahedral cage **5** by self-assembly based on pyridine ligands **4** and [*cis*-(en)Pd(NO_3_)_2_] **3** (Figure 2) [82]. Significantly, the [*cis*-(en)Pd(NO_3_)_2_] unit, prepared from starting materials ethylenediamine **1** and PdCl_2_, was crucial for successful self-assembly (Figure 2). The authors indicated that the cationic charge carried by the palladium (II) centers of the octahedron **5** is the key to enhanced hydrophilicity, although the cage has a large hydrophobic cavity. On the other hand, the multiple [(en)Pd]^2+^ corners may be regarded as solubilizing groups due to ethylenediamine being highly soluble in water. In addition, the nitrate counterions also enhance the hydrophilicity of octahedron **5** compared to chloride ions [83,84]. So, the [*cis*-(en)Pd(NO_3_)_2_] units could provide structural stability and water solubility. As shown in Figure 2, various WSMOCs have been synthesized by self-assembly of the water-soluble [(en)Pd]^2+^ or [(en)Pt]^2+^ corners with different pyridine-based ligands **6**–**11** in recent years [85,86].

### 2.2. Charged Ligands

As shown in Figure 3a, the Nitschke group reported a new type of water-soluble tetrahedral cage **14**, which was constructed by a charged ligand through sub-component self-assembly in water [87]. The subcomponents 2-formylpyridine **12** and 4,4’-diaminobiphenyl-2,2’-disulfonic acid **13** containing two solubilizing sulfonate groups were assembled with iron(II) in an aqueous solution of tetramethylammonium hydroxide to afford tetrahedral cage **14** in nearly quantitative yields. Cage **14** has high hydrophilicity, mainly because 12 highly hydrophilic sulfonate groups extend out of the cavity and are arranged symmetrically through the crystal structure analysis. It is noted that the hydrophobic internal cavity (141 Å^3^) can encapsulate a wide range of guests [88,89,90,91]. This method has been shown to be useful for the construction of water-soluble cages with charged properties. For example, cages **16**, **19**, and **21** were prepared from ligands that are anchored with ammonium **15** and pyridinium **17** and assembled from ligand **20** after dehydrogenation under alkali conditions, respectively (Figure 3b–d) [92,93,94].

### 2.3. Hydrophilic Groups

The construction of WSMOCs could rely on the synthetic modification of water-insoluble metallocages with neutral hydrophilic groups on the skeletons (Figure 4). For example, Nitschke and co-workers reported chiral WSMOC *ΔΔΔΔ*-**23**/*ΛΛΛΛ*-**23** assembled from iron(II) sulfate with a chiral ligand **22** containing two chiral glyceryl groups and 2-formylpyridine **12** in water [95]. It is noteworthy that the glycerol groups not only enhance the water solubility of the cage *ΔΔΔΔ*-**23**/*ΛΛΛΛ*-**23** but also determine the handedness of the iron(II) stereocenters. In addition to this, to enhance hydrophilicity of metallo-supramolecular cages, other neutral hydrophilic groups can also be introduced into the skeletons, such as methoxyethoxy, glucose, polyethylene glycol (PEG), etc. (Figure 4b–d) [96,97,98,99,100,101,102].

### 2.4. Anion Exchange

Sometimes it is difficult to directly form a WSMOC through the methods above. In 2017, Nitschke and co-workers established another method for constructing WSMOC **31** (Figure 5) [103]. First, they attempted to prepare cube **31** by direct self-assembly of subcomponents zinc(II)-porphyrin **30** and 2-formylpyridine **12** with ferrous (II) sulfate in different solvents and were unsuccessful, as the results were suspensions that were insoluble in the assembled solvents. Then, they used Fe(OTf)_2_ instead of FeSO_4_ assembled with other components to successfully obtain cube **32**, which is basically insoluble in water. Finally, the anion exchange of cage **32** with tetrabutylammonium (TBA) sulfate in MeCN induces the spontaneous precipitation of the sulfate salt of cube **31**, which can be easily dissolved in water. The anion-exchange strategy is an effective way to synthesize WSMOCs from highly hydrophobic ligands [104,105].

## 3. Photocatalysis in WSMOCs

The WSCCs with controlled size, shape, and hydrophobic cavities have been further used in catalysis [98,106], guest encapsulation [107,108], drug delivery [109], light harvesting [110], sensing [111,112,113], and photoswitching [114,115,116]. In recent years, the application of metal-organic cages in light-driven catalysis has been very fruitful. The photosensitizer (PS) generally has two different productive mechanisms after excitation (Figure 6a). Firstly, the excited state (PS*) transfers energy to the energy acceptor ([A]) and then forms a new excited state ([A]*). Another productive mechanism is that an oxidized (PS^+^) or reduced (PS^−^) species is first formed by a photoinduced electron transfer (PET) process, followed by single electron transfer (SET) back to the original state. Crucially, both significantly rely on the distance between the energy donor and acceptor, which can be controlled by the preorganization of metal-organic cages. The photosensitizer can achieve different types of photocatalytic reactions in water by binding to different positions of the metal-organic cages (Figure 6b).

Since the first host–guest binding was reported, one of these applications is the ability of WSMOCs to catalyze photoreactions, which can mimic artificial photosynthesis [117,118]. Great strides have been made toward achieving this goal, although photochemical ones are notoriously difficult to control due to their intrinsically low reaction barriers upon excitation and highly reactive intermediates. In recent years, REDOX reactions and C-H activation reactions under mild conditions have been frequently reported, but it is not easy to realize these reactions in the water phase [119,120,121,122]. Herein, we focus on the application of WSMOCs for photocatalysis in artificial photosynthesis and in organic photo(redox) catalysis. It is worth noting that the hydrophobic cavity of WSMOCs realizes the preorganization of the reaction substrate by the hydrophobic effect, which is the key to promote the reactions in water (Figure 7). In our discussion, depending on the different ways by which WSMOCs can mediate photocatalytic reactions, this will be covered in the following four sections: (1) photoredox catalysis mediated by the cavity of WSMOCs; (2) photoredox catalysis mediated by the ligand or metal of WSMOCs; (3) photoredox catalysis mediated by the WSMOC-based light-harvesting system.

### 3.1. Photoredox Catalysis Mediated by the Cavity of WSMOCs

Various guests were inserted into the unique cavities of WSMOCs by means of hydrophobic effects, π-π interactions, electrostatic interaction, and van der Waals interactions [123,124,125]. Therefore, the substrates can be encapsulated in the microenvironments of WSMOCs and undergo photoredox reactions inside the cavities.

In 2002, the group of Fujita reported that the structurally well-defined coordination cages act as nano-reactors to promote intermolecular [2 + 2] photodimerization of large olefins in a surprisingly efficient fashion and control the stereo- and regiochemistry in a stringent geometrical environment (Figure 8a) [126]. It was found that the internal cavity of cage **5b** can bind two molecules of acenaphthylene **33**, and then the syn-dimer was obtained at a near quantitative yield in water under light irradiation, without any other regions or stereoisomers being produced. However, in the absence of **5b**, the yield of the mixture of multiple dimer isomers obtained was very low, and even the concentrations of substrate **33** were very high. The cages were readily obtained, and their unique cavities make it possible to create new chemical reactions within the localized microenvironment of discrete molecules. For example, under visible light, cage **5** can selectively encapsulate two different reaction substrates and then convert them into corresponding dimer products in quantitative yields. Among the reactions, most of them focused on [2 + 4] or [2 + 2] cycloadditions of arenes and inert aromatics **34**–**38** (Figure 8b–e) [127,128,129,130], cyclization of α-diketones **39** (Figure 8f) [131], and 1,4-radical addition of compounds **40** and **41** (Figure 8g) [132]. However, in these photoreactions, cages **5a** or **5b** only provide defined microenvironments to control the photo-reactions and do not take part in the photocatalytic process.

Moreover, the host–guest complexation is essential to the photooxidation [133,134]. The cage and guest are often stably bound together by non-covalent interactions, which may result in a charge transfer (CT) process that leads to new absorption bands that lie within the visible light energy. These absorption bands may undergo a single electron transfer (SET) process from donor to acceptor upon photoirradiation, creating a radical pair. However, these radical pairs are extremely unstable, hampering their use in diffusion-dependent chemistry [135]. Therefore, photochemical ones are difficult to control due to their intrinsically low reaction barriers upon excitation and highly reactive intermediates. As a result, it is critical to develop a host–guest complex for stabilizing reactive intermediates.

The group of Sun reported a series of water-soluble redox-active WSMOCs [136,137]. Cage **43** can be easily synthesized by self-assembly of ligand **42** and palladium salts without guests; however, cage **44** is formed in the presence of a guest. It is worth noting that the transformation between the two structures can be effectively controlled through the addition and release of guests (Figure 9a). Compared to the cage designed by Fujita, this cage also provides reversible redox-activities, and it is bigger with bigger internal cavities that can accommodate more guests, such as aromatic molecules and polyoxometalates (POMs). Moreover, POMs@Pd_4_L_2_ host–guest complexes can enhance the conversion and product selectivity in the desulfurization reaction of aryl sulfide. Control experiments showed that the cage, air, and light were essential for the reaction. As shown in Figure 9b, they proposed the mechanism for the desulfurization reaction. Firstly, cage **43** is transformed into bowl cage **44** after encapsulation of aryl sulfide **45**. The resulting host–guest complex undergoes effective electron transfer after excitation by visible light to produce superoxide anion, thus oxidizing sulfide to sulfoxide. Finally, sulfoxide **46** is released by the cage due to it being highly hydrophilic, causing cage **43** to regain it. In conclusion, they designed a guest-controlled enzyme simulation photocatalytic system for the oxidation of aryl sulfide.

Recently, the same group developed a new WSMOC **48**, which was prepared from pyridinium-bonded macrocyclic ligand **47** and [*cis*-(en)Pd(NO_3_)_2_] (Figure 10) [138]. Under the same assembly conditions, the host−guest complex [**49**⊂**48**] can be synthesized in the absence of decatungstate **49**. The host–guest complex [**49**⊂**48**] formed by the installation of the electron-rich guest **49** was encapsulated by the cationic host with REDOX activity via electrostatic interaction and is an ideal platform for green catalysis because it can efficiently promote the C−H photooxidation of toluene derivatives **50** to aldehyde products **51** with good yields and selectivity in water. Compared to adding a photocatalyst alone, such a host−guest complex [**49**⊂**48**] showed enhanced catalytic activity in catalyzing the C−H photooxidation. It is noteworthy that the co-encapsulation of catalyst and substrate by electrostatic interaction and hydrophobic action in the hydrophobic cavity of WSMOC **48** is the key to promoting the reaction.

### 3.2. Photoredox Catalysis Mediated by the Ligand or Metal of WSMOC

The design of WSMOCs, which possess light-absorbing functionalities, is interesting for molecular sensing and photocatalysis. The key to obtain a light-absorbing WSMOC is to select the desired organic ligand. The group of Fujita reported the first example of a photocatalytic reaction driven by a self-assembled cage **5a** as a photosensitizer [139]. The cage **5a** showed strong absorption at <370 nm, and the ligand should behave as a strong electron acceptor due to electron donation to Pd^2+^ centers at three pyridine coordination sites. In 2019, they found that the demethylenation reactions of cyclopropanes **53** can be mediated within the cavity of cage **5a** under visible light irradiation (Figure 11a) [140]. They suggest a plausible mechanism for this reaction: (a) The reaction proceeds via a guest-to-host electron transfer (ET) because of cage **5a** being a strong electron acceptor, and then gives a radical cation of a cyclopropane substrate; (b) under the nucleophilic addition of a nitrate anion, the ring of the radical cationic cyclopropane opens to form a radical intermediate; (c) the radical intermediate is broken to form the demethylenated product and nitrite radical; and (d) the nitrate gets one electron from the anion radical of cage **5a** to complete the reaction. Therefore, the reactions were highly chemo-selective and enabled late-stage derivatization of a steroid molecule, which led to a totally new unnatural steroid. Similarly, cage **5a** can also be used for mediating anti-Markovnikov hydration of aryl alkynes **55** (Figure 11b) [141] and photo-oxidation reaction of triquinacene **56** (Figure 11c) [142].

The water-soluble tetrahedral cage **57**, designed by the group of Raymond, also has visible light absorption (Figure 12) [143]. Fortunately, WSMOC 57 can act as a photosensitizer and photosensitizes an allylic 1,3-rearrangement of encapsulated cinnamyl ammonium substrate **58**. They suggest a plausible mechanism for this reaction by UV/vis absorption, fluorescence, ultrafast transient absorption, and electrochemical experiments: (a) This cage easily encapsulates a cinnamyl ammonium substrate **58** to generate the excited charge-transfer state **58**⊂**57***, in which an electron has been donated to **58** after absorbing the energy of visible light; (b) After gaining an electron from the cage, the C−N bond of the substrate is broken to form a tertiary amine and a geminal radical ion pair; (c) The electron of the allyl or tertiary amine back transfers to the ligand-based radical cation and then forms a stable allyl cation or tertiary amine radical cation and reestablishes the original charge on the ligand; (d) The encapsulated tertiary amine recombines with the allyl cation within the cavity to form the 1,3-rearrangement product. In summary, this process is a photoinduced electron transfer (PET) mechanism proven by photochemical and photophysical processes.

The Su’s group designed a WSMOC **60** prepared from Ru(II)-based photoactive ligand and palladium(II) salt in a quantitative yield (Figure 13) [144]. The octahedral cage **58** can easily provide ET processes and electron collection pathways, including intra-ligand charge transfer (ILCT); ligand-to-metal charge transfer (IMCT); and metal-ligand charge transfer (MLCT). They found that the homochiral *ΔΔΔΔ*-**60**/*ΛΛΛΛ**-*****60** can encapsulate naphthol substrates and undergo an asymmetric 1,4-coupling reaction of **61** to form dimerization product **62** with high stereoselectivity under visible light. The homochiral **58** could control the stereoselectivity of the product (Figure 13a, left). They give a radical reaction mechanism by various techniques, including ESR spectroscopy, CV, and UV/vis absorption and emission spectroscopy, as well as various control experiments (Figure 13b). The photoredox Ru^II^ centers are irradiated by visible light to produce a *[**60**]^2+^ excited state, which is quenched by O_2_ to give [**60**]^3+^. The [**60**]^3+^ may then oxidize the naphthol substrate **61** by single-electron transfer to produce radical species A, which reacts with a hydroxyl radical to produce the intermediate B. Afterwards, egioselective 1,4-coupling occurs exclusively to produce **62** without detectable amounts of 1,1’-bi-2-naphthol. Recently, the same group reported that the [2 + 2] photocycloaddition of acenaphthylene **63** can also be asymmetrically photo-catalyzed by homochiral cage **60** (Figure 13a, right) [145]. At about the same time, they also reported that **60** can achieve the intermolecular [2 + 2] cycloaddition reaction of *α,β-*unsaturated carbonyl compounds with high selectivity under blue light irradiation (Figure 13b) [146]. This intermolecular [2 + 2] cycloaddition reaction can also be mediated by the anthraquinone-based metal–organic cages, which was reported by the group of Cui [147]. This WSMOC has high stability and catalytic performance and can catalyze various substituent substrates with excellent yields. In addition, the host–guest complexation of this cage and tetrathiafulvalene (TTF) guests can improve the catalytic efficiency of visible-light-driven H_2_ evolution [148]. In these reactions, the ligands with photoactive centers and catalytic Pd metal centers play decisive roles in photooxidation and ET processes.

### 3.3. Photoredox Catalysis Mediated by the WSMOC-Based Light-Harvesting System

There is great potential to design artificial light-harvesting systems (LHSs), which can simulate enzyme catalysis in a water medium for effective application [149,150]. Fluorescence resonance energy transfer (FRET) is commonly used in the development of artificial light-harvesting systems (LHSs) and is essential for simulating natural photosynthesis processes [151,152,153]. Compared to the supramolecular complexes that rely on host–guest or hydrogen bond interaction, the FRET process is easier to perform in MOCs that are based on metal-ligand interaction, because it often needs to occur at very low concentrations of solution. However, the interactions of the former supramolecular complexes can easily disappear in a dilute solution, which makes them difficult to be used in the development of a light collection system [154,155,156].

In 2019, the group of Zhang developed a new WSMOC **68**, which was prepared from tetraphenylethene (TPE) skeleton **65** and polyethylene glycol (PEG) unit **66** [157]. The high-emission cage can be further self-assembled in water to form nanoaggregates. Then, the cage acts as an energy donor to combine with eosin Y to construct an LHS through the FRET process. More importantly, under visible light, this LHS can effectively mediate the cross-coupling reaction of benzothiazole **69** and diphenylphosphine oxide **70** in water (Figure 14a). Compared to eosin Y alone, this system showed enhanced catalytic activity in catalyzing the cross-coupling reaction because the FRET process increases the number of excited photocatalysts (eosin Y*), which undergo the catalytic cycle to increase the catalytic activity (Figure 14b).

Recently, Mukherjee and co-workers designed and constructed a series of tetragonal prismatic metallacages based on the modification of tetraphenylethene (TPE) unit **72** [158]. Compared with the dilute solution state, cage **74** has higher emissivity in both aggregate and solid states, which is due to the presence of the TPE skeleton with AIE activity and the increased quantum yield (Figure 15). Cage **74** can aggregate in 90% H_2_O/MeCN (9/1, *v*/*v*) to form a spherical nanoaggregate, which can act as an LHS to transfer energy to photosensitizer rhodamine B (RhB). It is worth noting that the light-harvesting material (**74** + RhB) can promote the cross-coupling cyclization of *N*,*N*-dimethylaniline **75** and *N*-alkyl/aryl maleimides **76** more effectively than the addition of RhB or cage **74** alone. The FRET process increases the number of excited photocatalysts (RhB*) and is the key to increasing the catalytic activity (Figure 15).

## 4. Conclusions and Outlook

As described above, WSMOCs have great potential applications in light-driven catalysis due to the defined size and shape of their hydrophobic cavities. They can also be used as green nanoreactors and are suitable for substrate preorganization, thus improving the region-selective and stereo-selective aspects of the products. Supramolecular self-assembly is a powerful strategy to build similarly complex and well-organized structures from simple building blocks, thus mimicking nature’s principles. To enhance the hydrophilicity of metallo-supramolecular cages, several synthetic strategies have been applied: (1) self-assembly based on small pyridyl panels with end-capped [cis-(en)Pd(NO3)2]; (2) self-assembly based on ligands that are anchored with anions or cations; (3) self-assembly based on ligands with hydrophilic groups; and (4) changing the counteranions to more water-soluble ones. We believe that the design strategies of WSMOCs can be extended to the construction of other water-soluble coordination supramolecules and stimulate the development of more and more new water-soluble coordination supramolecules, such as water-soluble metal rings, coordination polymers, chorohydrocarbons, rotane, etc.

Subsequently, we have shown that WSMOCs can be used as green nanoreactors to mediate photocatalytic reactions. Based on literature reports, we reasoned that encapsulating more types of catalytic species with various sizes and functional groups could further advance supramolecular photo-catalysis. However, implementing such a strategy would need to fulfill the following three criteria: (1) sufficiently large inner space of the WSMOC to accommodate structurally complicated catalysts and substrates, (2) sufficiently strong driving force to ensure encapsulation, and (3) a WSMOC with light absorption function, as it is more conducive to its application in the field of molecular sensing and photocatalysis.

Although great progress has been made in the design and synthesis of WSMOCs, the application of these cages in photocatalysis still needs to be further explored. For one thing, the types of WSMOCs used for photocatalysis are still scarce. New WSMOCs with two or more catalytically active sites need to be properly designed for a wider range of substrates. On the other hand, the photo-catalytic function of WSMOCs and the types of catalytic reactions still need to be developed. Furthermore, most WSMOCs have only one type of ligand. More types of catalytic reactions can be achieved by developing WSMOCs constructed with different ligands. This review provides inspiration for the future design of photocatalytic WSMOCs host-guest systems and their application in producing visible light and complex organic molecules and contributes to the development of green chemistry. A combination of photocatalytic metallacycles or metallacages with appropriate materials such as mesoporous carbon and some supramolecular hydrogels to construct robust hybrid materials is a feasible way to address this issue. Then the prepared hybrid materials can be immobilized to enable asymmetric catalysis with enhanced catalytic performance and better stability. Finally, the immobilization of supramolecular cages on a heterogeneous support like alumina or polymers may lead to recyclable and long-term stable photocatalysts based on coordination cages. In conclusion, the field of WSCCs as a sustainable development research area will continue to promote the development of host−guest chemistry; biological, environmental, and industrial applications; drug synthesis, etc. The synthesis of multifunctional WSMOCs will offer more pathways for using the output energy from the light-harvesting system to mimic the whole photosynthetic process.

## Figures and Tables

**Figure 1 molecules-28-04068-f001:**
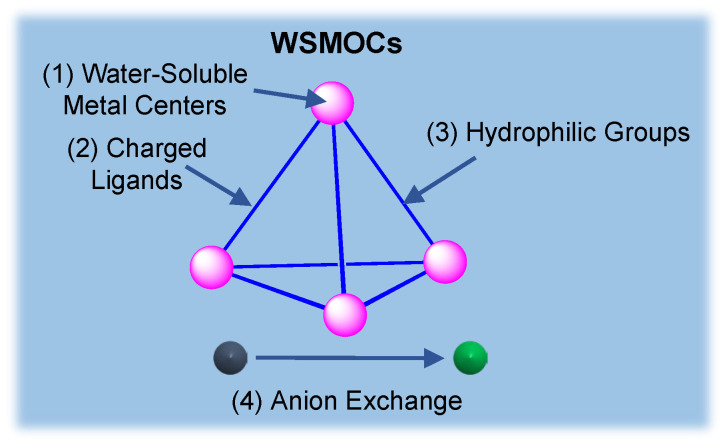
The scheme on the synthetic strategies of WSMOCs.

**Figure 2 molecules-28-04068-f002:**
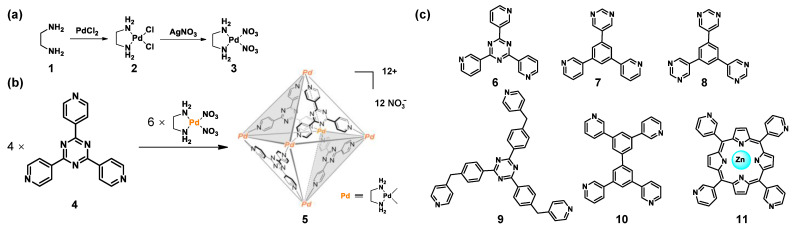
The synthesis of (**a**) [*cis*-(en)Pd(NO_3_)_2_]; (**b**) WSMOC 1; (**c**) different ligands used to synthesize WSMOCs.

**Figure 3 molecules-28-04068-f003:**
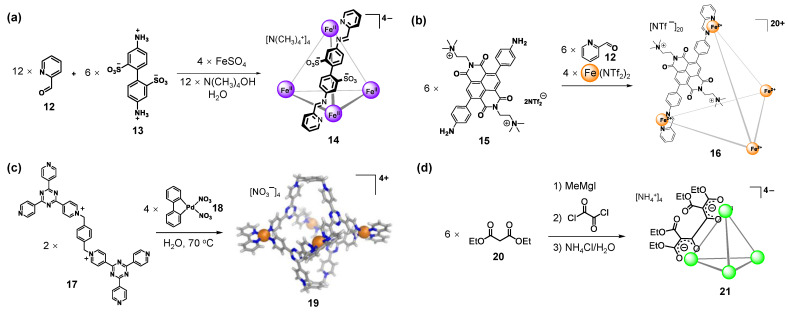
The reported water-soluble metallo-supramolecular cages assembled from ligands that are anchored with (**a**) sulfonate, (**b**) ammonium, and (**c**) pyridinium and (**d**) assembled from ligands after dehydrogenation under alkali conditions.

**Figure 4 molecules-28-04068-f004:**
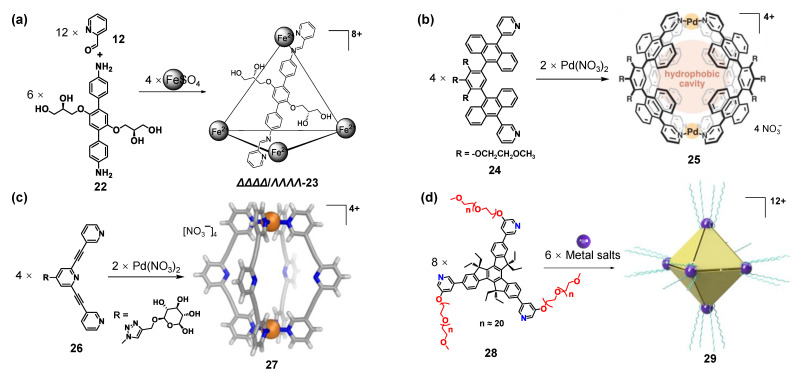
The reported water-soluble metallo-supramolecular cages assembled from ligands that are anchored with (**a**) chiral glyceryl, (**b**) methoxyethoxy, (**c**) glucose, and (**d**) polyethylene glycol.

**Figure 5 molecules-28-04068-f005:**
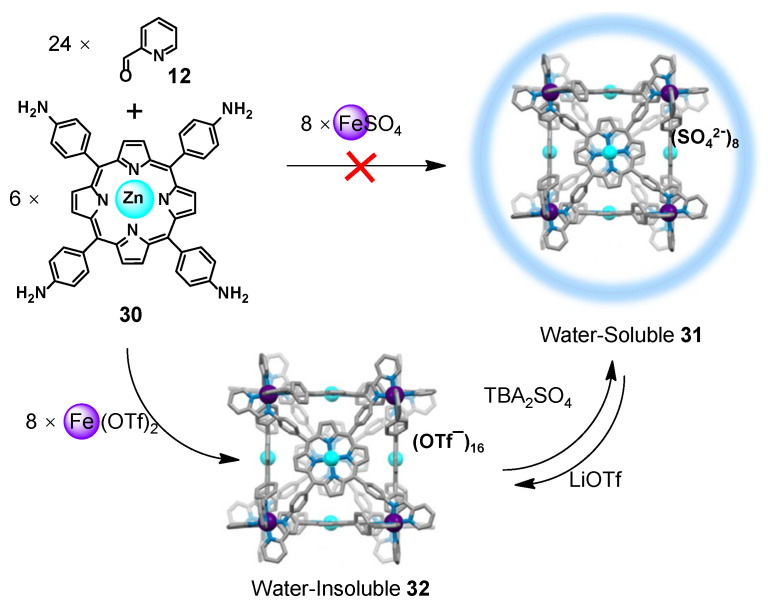
Preparation of water-insoluble cube **32** and water-soluble cube **31**.

**Figure 6 molecules-28-04068-f006:**
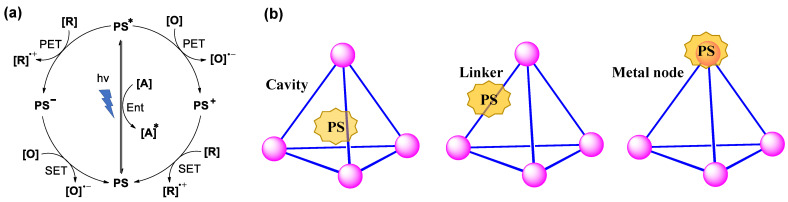
(**a**) General mechanisms of photo(redox)catalysis (Ent = energy transfer); (**b**) Schematic representation of metal-organic cages with the photosensitizer (PS) at different positions.

**Figure 7 molecules-28-04068-f007:**
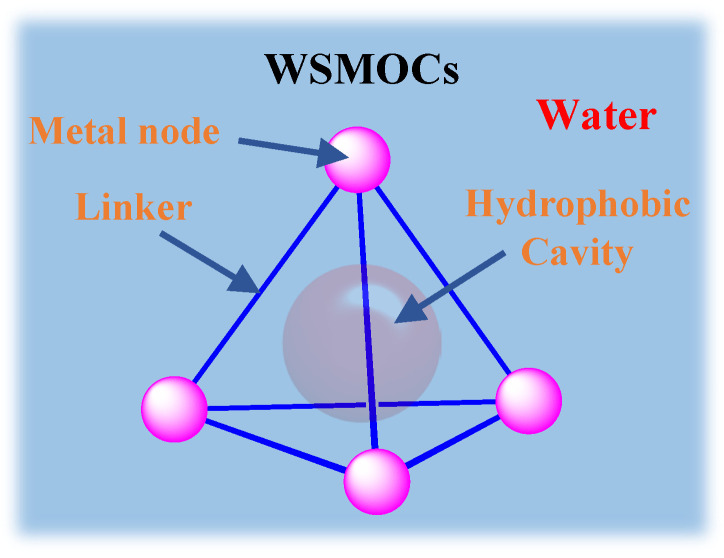
Schematic representation of a supramolecular coordination cage with different positions that are available for the introduction of (photo)catalytic functions.

**Figure 8 molecules-28-04068-f008:**
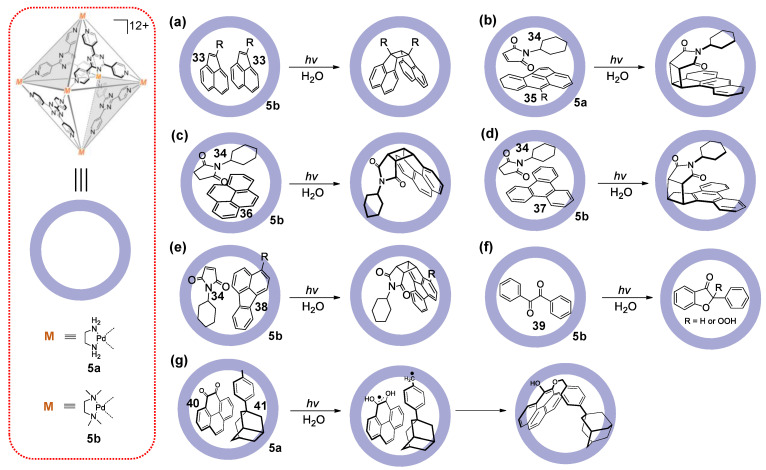
(**a**) The synthesis of ligand **L-A**; (**b**) The synthesis of ligand **L-B**; (**c**) The crystal structure of ligand **L-A**. (**d**–**g**) The synthetic applications.

**Figure 9 molecules-28-04068-f009:**
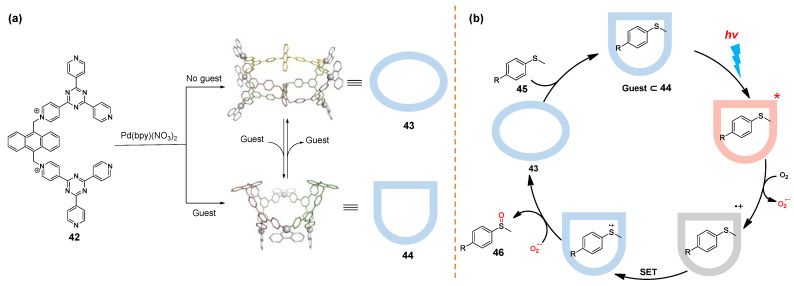
(**a**) Self-assembly of cages **43** and **44**; (**b**) Proposed mechanism of the photoinduced oxidation of sulfides to sulfoxides catalyzed by **43** and **44** hosts.

**Figure 10 molecules-28-04068-f010:**
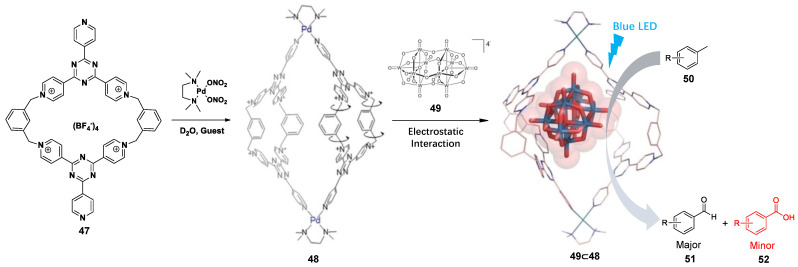
Self-assembly of cage **48** and host−guest complex [**49**⊂**48**]. Photocatalyzed aerobic oxidation of toluene derivatives with [**49**⊂**48**] under blue LED irradiation.

**Figure 11 molecules-28-04068-f011:**
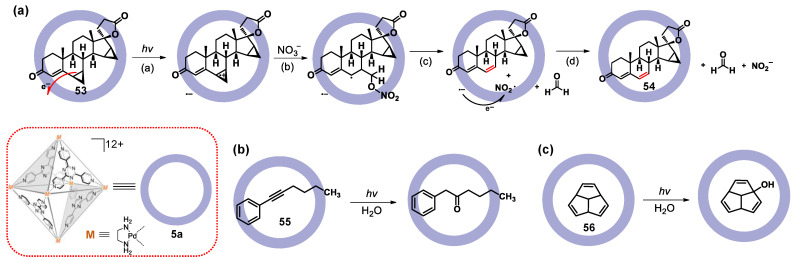
Examples showing (**a**) reaction mechanism of the demethylenation of **53**, (**b**) anti-Markovnikov hydration of aryl alkynes **55**, and (**c**) photo-oxidation reaction of triquinacene **56** via photoinduced guest-to-host electron transfer in cage **5a** under UV radiation.

**Figure 12 molecules-28-04068-f012:**
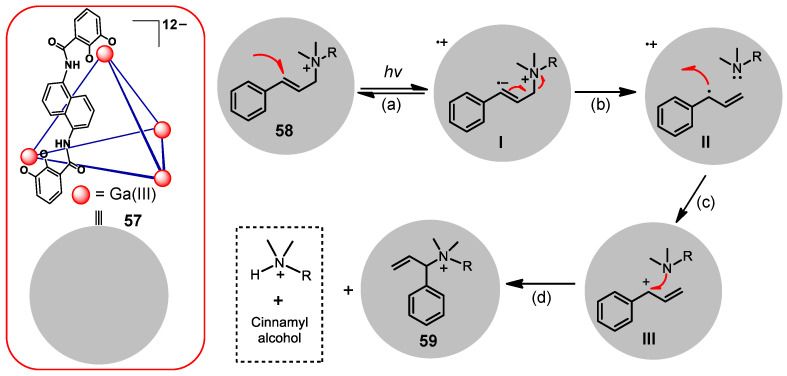
Example showing 1,3-rearrangment of **58** via photoinduced guest-to-host electron transfer in cage **57** under UV radiation.

**Figure 13 molecules-28-04068-f013:**
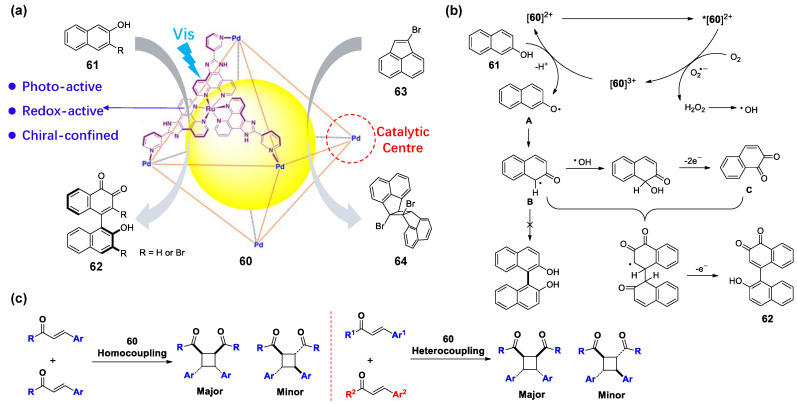
Examples showing (**a**) 1,4-coupling reaction of **61** and [2 + 2] photocycloaddition of acenaphthylene **63**; (**b**) possible reaction pathway of the oxidative 1,4-coupling; (**c**) [2 + 2] cycloaddition reaction of α,β-unsaturated carbonyl compounds via photoinduced guest-to-host electron transfer in cage **60** under UV radiation.

**Figure 14 molecules-28-04068-f014:**
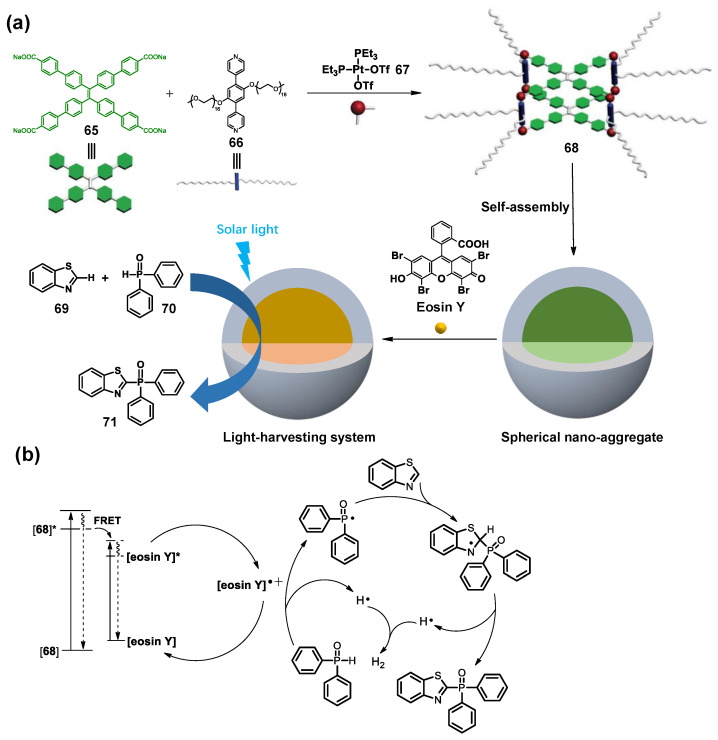
(**a**) Self-assembly of the WSMOC **68** and cartoon representations of the light-harvesting system and its application in a photocatalytic reaction; (**b**) Plausible mechanism of cross-coupling hydrogen evolution reaction using LHS as a photocatalyst.

**Figure 15 molecules-28-04068-f015:**
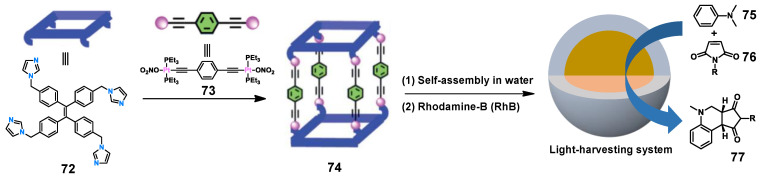
Self-assembly of the WSMOC **74** and cartoon representations of the light-harvesting system and its application in a photocatalytic reaction.

## Data Availability

Not applicable.

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
