# Peer review of "Photocatalysis in Water-Soluble Supramolecular Metal Organic Complex"

_molecules, 2023, doi:10.3390/molecules28104068_

Round 1
Reviewer 1 Report
In this review, the strategies for synthesizing water-soluble metal organic cages (WSMOCs) by altering metals and ligands are discussed, with a focus on their photoredox catalytic applications using WSMOC-based light-harvesting systems, cavity, ligands, and metal centers. The recent reviews have discussed the synthesis and potential applications of WSMOCs (e.g., Chem. Rev. 2020, 120, 24, 13480–13544, Chem. Commun., 2022,58, 5558-5573); this review explicitly explores the fascinating and promising topic of photocatalysis.
The review briefly touches upon the topic of WSMOCs for photocatalysis, it lacks a comprehensive analysis of the subject matter. The study requires providing a historical overview and evolution of metal-organic cages for photocatalysis, which is essential for readers to grasp the significance of the research.
The manuscript needs to thoroughly discuss the photocatalysis mechanism for WSMOCs, including their performance, rate, conversions, and how they influence photocatalysis. The study also lacks a rational and specific explanation of the type of ligands, metal coordination centers, and pore sizes required for photocatalysis and their influence on bandgap and photocatalytic performances.
It is important to note that presenting data from individual articles requires appropriate approvals from respective journals. However, the manuscript needs to acknowledge this essential aspect. In addition, the review does not include any WSMOC and its photocatalytic comparison tables, which could help readers understand the topic better.
This manuscript lacks a comprehensive analysis of WSMOCs for photocatalysis and appears more of a report than a review. Addressing these concerns could improve the manuscript and make it a valuable contribution to the literature on the subject.
Author Response
Response Letter
Prof. Milena Mirkovic,
Editor, Molecules
Dear Prof.Milena Mirkovic,
Thank you and the two reviewers for reviewing our paper entitled “Photocatalysis in Water-Soluble Supramolecular Metal Organic Complex” (Manuscript ID: molecules-2323820). We have revised the manuscript according to the comments and suggestions. As per the editor and reviewer’s comments, we have revised our manuscript and the detailed responses are given below with blue color. The changes made based on comments in the revised manuscript are highlighted in a yellow background. We believe these changes have resulted in a much more effective article that will be more suitable for being published in Molecules. Attached are the point-by-point responses to the reviewers’ comments. We hope the revised manuscript along with our response letter would be satisfactory.
Again, thank you for considering our manuscript and offering us the chance for revision. We look forward to seeing the acceptance of our manuscript soon.
Sincerely,
Linlin Shi
Point-by-Point Responses to Reviewers’ Comments
Reviewer #1:
In this review, the strategies for synthesizing water-soluble metal organic cages (WSMOCs) by altering metals and ligands are discussed, with a focus on their photoredox catalytic applications using WSMOC-based light-harvesting systems, cavity, ligands, and metal centers. The recent reviews have discussed the synthesis and potential applications of WSMOCs (e.g., Chem. Rev. 2020, 120, 24, 13480-13544, Chem. Commun., 2022,58, 5558-5573); this review explicitly explores the fascinating and promising topic of photocatalysis.
- The review briefly touches upon the topic of WSMOCs for photocatalysis, it lacks a comprehensive analysis of the subject matter. The study requires providing a historical overview and evolution of metal-organic cages for photocatalysis, which is essential for readers to grasp the significance of the research.
Response: We thank the reviewer for the encouraging comments. According to the constructive comments of reviewers, we have given a historical overview and a brief introduction of metal-organic cages for photocatalysis in the revised manuscript (please see page 6 for details, yellow background).
- The manuscript needs to thoroughly discuss the photocatalysis mechanism for WSMOCs, including their performance, rate, conversions, and how they influence photocatalysis. The study also lacks a rational and specific explanation of the type of ligands, metal coordination centers, and pore sizes required for photocatalysis and their influence on bandgap and photocatalytic performances.
Response: We highly appreciate this reviewer’s encouraging comments. According to the constructive comments of reviewers, we have made a further study on the mechanism in the revised manuscript.
- It is important to note that presenting data from individual articles requires appropriate approvals from respective journals. However, the manuscript needs to acknowledge this essential aspect. In addition, the review does not include any WSMOC and its photocatalytic comparison tables, which could help readers understand the topic better.
Response: We thank the reviewer for the encouraging comments. Although strategies for the synthesis of WSMOCs have been reported, there are few reviews on photocatalytic reactions of these cages in aqueous phase. According to your suggestion, we have made a further analysis of WSMOCs for photocatalysis in the revised manuscript to help readers understand the topic better.
- This manuscript lacks a comprehensive analysis of WSMOCs for photocatalysis and appears more of a report than a review. Addressing these concerns could improve the manuscript and make it a valuable contribution to the literature on the subject.
Response: We appreciate the reviewer’s comments. As suggested, we have given the analysis of WSMOCs for photocatalysis in the revised manuscript.
Reviewer #2:
The manuscript by D. Hong et al describes on “Photocatalysis in Water-Soluble Supramolecular Metal Organic Complex”. Here, the authors wrote an excellent review article on the topic. A large number of cavity-containing supramolecules, such as metal-organic cages (MOCs), have been extensively explored for a large variety of reactions with a high degree of reactivity and selectivity. Because sunlight and water are necessary for the process of photosynthesis, water-soluble metal-organic cages (WSMOCs) are ideal platforms for photo-responsive stimulation and photo-mediated transformation by simulating photosynthesis due to their defined sizes, shapes, and high modularization of metal centers and ligands. After a nice introduction on the topic, they discussed the synthesis of this class of WSMOCs that include metal centers, charged ligands, hydrophilic groups, and anion exchange methods. After the synthesis, they discussed three (not four) photocatalytic reactions mediated by WSMOCs. They include photoredox catalysis mediated by the cavity of WSMOC; photoredox catalysis mediated by the ligand or metal of WSMOC; and photoredox catalysis mediated by the WSMOC-based light-harvesting system. Finally, they discussed on the future applications of this class of materials. It contained nicely drawn 12 Figures and 149 references (good number of references, given the length of this manuscript). Therefore, I recommend its publication in molecules as its present form.
Response: We thank this reviewer for the valuable concern about our present work and for accepting our manuscript. We have revised the manuscript in order to help more readers to understand the topic, which highlighted in a yellow background.
Reviewer #3:
Photoactive materials have become extremely popular in the large variety of applications in the fields of photocatalytic degradation of pollutants, water splitting, organic synthesis and etc. In this review, methods for the construction of WSMOCs and its application in photocatalysis are shown. In the review some examples with figures by synthesis are taken from another review Chem. Rev. 2020, 120, 24, 13480–13544 (https://doi.org/10.1021/acs.chemrev.0c00672). For example: fig.1 (this review) corresponds to fig. 4 and fig. 7 (from hem. Rev. 2020, 120, 24, 13480–13544); fig 2 (this review) - fig. 8, fig. 15, fig. 2 (from hem. Rev. 2020, 120, 24, 13480–13544); fig. 3 (this review) - fig. 9, fig. 11, fig. 13 (from hem. Rev. 2020, 120, 24, 13480–13544); fig. 4 (this review) - fig. 16 (from hem. Rev. 2020, 120, 24, 13480–13544); fig. 4 (this review) - fig. 16 (from hem. Rev. 2020, 120, 24, 13480–13544); fig. 5 (this review) - fig. 76 (from hem. Rev. 2020, 120, 24, 13480–13544). I think authors should be focus on a topic in application or describe new methods.
Response: We thank the reviewer for the encouraging comments. In recent years, although there have been many reports on metal-organic cages mediated photocatalytic reactions, most of them are carried out in organic solvents. Since the first host-guest binding was reported, one of these applications is the ability of WSMOCs to catalyze photoreactions, which can mimic artificial photosynthesis. Great strides have been made toward achieving this goal, although photochem-ical ones are notoriously difficult to control due to their intrinsically low reaction barriers upon excitation and highly reactive intermediates. Herein we focus on the application of WSMOCs for photocatalysis in artificial photosynthesis and in organic photo(redox) ca-talysis. It is worth noting that the hydrophobic cavity of WSMOC realizes the preorganization of the reaction sub-strate by hydrophobic effect, which is the key to promote the reactions in water (Figures 6 and 7). We believe that this review will help more readers to understand the photocatalytic reactions mediated by WSMOCs in aqueous media, which is also in line with the concept of green chemistry development.
Reviewer #4:
- The manuscript contains spelling/grammatical errors. So, the language should be polished thoroughly.
Response: We thank the reviewer for the encouraging comments. Addition to the reviewers’ comments, we also carefully polished the language in the revised manuscript. All the revisions have been highlighted.
- I suggest the authors could give a scheme on the synthetic methods or strategy.
Response: We thank the reviewer for the encouraging comments. In the revised manuscript, we have given a scheme on the synthetic strategies, which are depicted in Figure 1.
- 3.What potential does further research hold? What is the ultimate goal in this field?
Response: We thank the reviewer for the encouraging comments. In this review, strategies are presented that address key challenges for the preparation of coordination cages that are soluble and stable in water. Herein we focus on the application of supramolecular cages for photocatalysis in artificial photosynthesis and in organic photo(redox) catalysis. Then, a brief overview of immobilization strategies for supramolecular cages provides tools for implementing cages into devices. This review provides inspiration for future design of photocatalytic WSMOCs host-guest systems and their application in producing visible light and complex organic molecules, and contributes to the development of green chemistry.
- 4.Does the future of study lie in this area? Are there other more promising areas in the field which could be progressed?
Response: We thank the reviewer for the encouraging comments. From a chemical perspective, excited states generated by light make thermodynamically uphill reactions possible, which forms the basis for energy storage into fuels. In addition, with light, open-shell species can be generated which open up new reaction pathways in organic synthesis. Crucial are photosensitizers, which absorb light and transfer energy to substrates by various mechanisms, processes that highly depend on the distance between the molecules involved. Supramolecular coordination cages, especially water-soluble metal-organic cages (WSMOCs) with unique hydrophobic cavities, are well studied and synthetically accessible reaction vessels with single cavities for guest binding, ensuring close proximity of different components. Due to high modularity of their size, shape, and the nature of metal centers and ligands, cages are ideal platforms to exploit preorganization in photocatalysis. In addition to this, the field of WSCCs as a sustainable development research area will contin-ue to promote the development of host-guest chemistry, biological, environmental, indus-trial, and drug synthesis, etc.
- 5.In the introduction, I suggest these authors should discuss detailed about the difference of this material with other materials.
Response: We thank this reviewer for the constructive comment. We have discussed detailed about the difference of this material with other materials in the introduction, which highlighted in a yellow background. Hopefully the revised language would be satisfactory.
- 6.Some related work on the design synthesis could be cited, including Org. Chem. Front., 2020,7, 3515-3520; Chem. Commun., 2022, 58, 6653–6656; Molecules, 2019, 24, 1760 and J. Org. Chem. 2019, 84, 14627−14635.
Response: We appreciate the reviewer comments. As suggested, we have cited the mentioned references in the revised manuscript.
- 7.The section of Conclusion and outlooks is too simple, the authors should give deeper insights into the advantages, loopholes and future development direction of complexes.
Response: We thank the reviewer for this comment. We have revised the section of conclusion and outlooks, which highlighted in a yellow background. Hopefully the revised language would be satisfactory.
Finally, we thank all the reviewers again for their critical and constructive comments.
Reviewer 2 Report
The manuscript by D. Hong et al describes on “Photocatalysis in Water-Soluble Supramolecular Metal Organic Complex”. Here, the authors wrote an excellent review article on the topic. A large number of cavity-containing supramolecules, such as metal-organic cages (MOCs), have been extensively explored for a large variety of reactions with a high degree of reactivity and selectivity. Because sunlight and water are necessary for the process of photosynthesis, water-soluble metal-organic cages (WSMOCs) are ideal platforms for photo-responsive stimulation and photo-mediated transformation by simulating photosynthesis due to their defined sizes, shapes, and high modularization of metal centers and ligands. After a nice introduction on the topic, they discussed the synthesis of this class of WSMOCs that include metal centers, charged ligands, hydrophilic groups, and anion exchange methods. After the synthesis, they discussed three (not four) photocatalytic reactions mediated by WSMOCs. They include photoredox catalysis mediated by the cavity of WSMOC; photoredox catalysis mediated by the ligand or metal of WSMOC; and photoredox catalysis mediated by the WSMOC-based light-harvesting system. Finally, they discussed on the future applications of this class of materials. It contained nicely drawn 12 Figures and 149 references (good number of references, given the length of this manuscript). Therefore, I recommend its publication in molecules as its present form.
Author Response

(The authors gave the same response as above.)

Reviewer 3 Report
Photoactive materials have become extremely popular in the large variety of applications in the fields of photocatalytic degradation of pollutants, water splitting, organic synthesis and etc. In this review, methods for the construction of WSMOCs and its application in photocatalysis are shown. In the review some examples with figures by synthesis are taken from another review Chem. Rev. 2020, 120, 24, 13480–13544 (https://doi.org/10.1021/acs.chemrev.0c00672). For example: fig.1 (this review) corresponds to fig. 4 and fig. 7 (from hem. Rev. 2020, 120, 24, 13480–13544); fig 2 (this review) - fig. 8, fig. 15, fig. 2 (from hem. Rev. 2020, 120, 24, 13480–13544); fig. 3 (this review) - fig. 9, fig. 11, fig. 13 (from hem. Rev. 2020, 120, 24, 13480–13544); fig. 4 (this review) - fig. 16 (from hem. Rev. 2020, 120, 24, 13480–13544); fig. 4 (this review) - fig. 16 (from hem. Rev. 2020, 120, 24, 13480–13544); fig. 5 (this review) - fig. 76 (from hem. Rev. 2020, 120, 24, 13480–13544). I think authors should be focus on a topic in application or describe new methods.
Author Response

(The authors gave the same response as above.)

Reviewer 4 Report
1. The manuscript contains spelling/grammatical errors. So, the language should be polished thoroughly.
2. I suggest the authors could give a scheme on the synthetic methods or strategy.
3. What potential does further research hold? What is the ultimate goal in this field?
4. Does the future of study lie in this area? Are there other more promising areas in the field which could be progressed?
5. In the introduction, I suggest these authors should discuss detailed about the difference of this material with other materials.
6. Some related work on the design synthesis could be cited,including Org. Chem. Front., 2020,7, 3515-3520; Chem. Commun., 2022, 58, 6653–6656; Molecules, 2019, 24, 1760 and J. Org. Chem. 2019, 84, 14627−14635
7. The section of Conclusion and outlooks is too simple, the authors should give deeper insights into the advantages, loopholes and future development direction of complexes.
Author Response

(The authors gave the same response as above.)

Round 2
Reviewer 1 Report
After thorough revision and refinement, the manuscript has exhibited substantial improvement in terms of quality and coherence. Consequently, I am pleased to accept the manuscript for publication. The revisions made to the manuscript have contributed to a clearer and more concise presentation of the ideas, resulting in a more impactful and effective piece of scholarly work.
Reviewer 3 Report
After thorough revision, the manuscript has improved of quality . And i think the article can be accept for publication. The revisions are on the high-level
Reviewer 4 Report
accept